# Effect of utilizing either a self-reported questionnaire or administrative data alone or in combination on the findings of a randomized controlled trial of the long-term effects of antenatal corticosteroids

**Mohammad Shahbaz**[1], **Jane E. Harding**[1], **Barry Milne**[2], **Anthony Walters**[1], **Martin von Randow**[2], **Greg D. Gamble**[1]*, for the ANCHOR Study Group¶

1 Liggins Institute, The University of Auckland, Auckland, New Zealand, 2 Centre of Methods and Policy Application in Social Sciences, University of Auckland, Auckland, New Zealand

¶ Membership of the ANCHOR Study Group can be found in the Acknowledgments.
* gd.gamble@auckland.ac.nz

**Data Availability Statement:** Data cannot be shared publicly due to privacy concerns. We are

## Abstract

### Introduction

A combination of self-reported questionnaire and administrative data could potentially enhance ascertainment of outcomes and alleviate the limitations of both in follow up studies. However, it is uncertain how access to only one of these data sources to assess outcomes impact study findings. Therefore, this study aimed to determine whether the study findings would be altered if the outcomes were assessed by different data sources alone or in combination.

### Methods

At 50-year follow-up of participants in a randomized trial, we assessed the effect of antenatal betamethasone exposure on the diagnosis of diabetes, pre-diabetes, hyperlipidemia, hypertension, mental health disorders, and asthma using a self-reported questionnaire, administrative data, a combination of both, or any data source, with or without adjudication by an expert panel of five clinicians. Differences between relative risks derived from each data source were calculated using the Bland-Altman approach.

### Results

There were 424 participants (46% of those eligible, aged 49 years, SD 1, 50% male). There were no differences in study outcomes between participants exposed to betamethasone and those exposed to placebo when the outcomes were assessed using different data sources. When compared to the study findings determined using adjudicated outcomes, the mean difference (limits of agreement) in relative risks derived from other data sources were: self-reported questionnaires 0.02 (-0.35 to 0.40), administrative data 0.06 (-0.32 to 0.44),

obliged by the Northern A Health and Disability Ethics Committee of New Zealand to maintain the privacy of participants who consented to this research and to the national data sovereignty understanding of research involving New Zealand Indigenous peoples and by the Liggins Institute's own data sharing operating procedures (Liggins Institute Data Access Committee). For these reasons unrestricted access to data which might identify participants or investigate outcomes in indigenous peoples beyond the scope of the original ethics approval are not possible. However, De-identified participant data will be available to researchers who provide a methodologically sound proposal with appropriate ethical approval, where necessary, and following approval of the proposal by the Data Access Committee of the Liggins Institute. Data requestors will be required to sign a data access agreement before data are released. Request for access to data can be made to the Maternal and Perinatal Research Hub at the Liggins Institute, University of Auckland (Researchhub@auckland.ac.nz).

**Funding:** The 50-year follow-up study was funded by grant 19/690 from the Health Research Council of New Zealand. AW was funded in part through the Aotearoa Foundation (9909494). The work was also supported through the Auckland Medical Research Foundation (1421003, AW), and the Health Research Council of New Zealand (19/690, GG, BM, JH). These funding sources had no role in the data or manuscript preparation.

**Competing interests:** The authors have declared that no competing interests exist.

both questionnaire and administrative data 0.01 (-0.41 to 0.43), and any data source, 0.01 (-0.08 to 0.10).

## Conclusion

Utilizing a self-reported questionnaire, administrative data, both questionnaire and administrative data, or any of these sources for assessing study outcomes had no impact on the study findings compared with when study outcomes were assessed using adjudicated outcomes.

## Introduction

Data linkage to routinely collected data has numerous advantages for health researchers [1]. It can provide a high response rate, data on a range of outcomes in large studies, and comprehensive information on hard-to-reach sub-populations, all with less burden for participants and potentially lower costs for researchers compared to self-reported questionnaires [2–4]. Furthermore, data linkage enables the study of low-prevalence exposure-disease associations with comprehensive follow-up and continuous data collection [5], and it is an acceptable method among research ethics boards and patients [6]. One notable advantage of outcomes derived from routinely collected data is the implementation of a formal blinded outcome assessment [7]. This is because those responsible for collecting routine data are generally separate from the researchers making use of these data. Although blinded in-person assessments may enhance case ascertainment [8], this might not be as practical as data linkage in follow-up studies [9]. The potential reduction in costs with data linkage could also enhance the feasibility of extending clinical trial follow up [10], which may enable researchers to identify long-term benefits or harms of interventions [3], especially in countries where individuals can be uniquely identified by a national patient identifier.

However, using administrative datasets also presents some challenges. The misclassification of outcomes or the absence of information could diminish the accuracy of treatment effect estimations [11]. Clinical trials that utilize routinely collected data for determining outcomes commonly show smaller treatment benefits compared to traditional trials that do not rely on such routinely collected data [10]. Applying for data can be a difficult and time consuming process [12].

The self-reported questionnaire is an effective technique that also can provide reliable information about participants' clinical outcomes [13]. Several studies have found a high level of agreement between self-reported questionnaire data and administrative datasets for health outcomes [14, 15] However, the accuracy and reliability of self-reported data can be affected by many factors, including participants' capacity to recall diagnoses, their willingness to provide medical information, and the intricacy of the condition being reported [16–18]. Moreover, employing self-reported questionnaires can be resource-intensive and financially burdensome, particularly in the context of extensive research endeavours [19].

In principle, the combination of self-reported outcomes with administrative datasets could enhance outcome ascertainment and alleviate the limitations of both.

We have investigated the use of these combined data sources in undertaking a 50-year follow-up of a randomized trial of antenatal corticosteroids. Although corticosteroids were shown in that trial to reduce perinatal morbidity and mortality, animal studies have reported long-term adverse effects of antenatal corticosteroid exposure on the offspring, including

higher blood pressure in rats and sheep [20–22], and increased basal insulin-to-glucose ratio in sheep [23]. Human observational studies have also reported that children exposed to antenatal corticosteroids had higher blood pressures and a higher incidence of mental disorders compared to those not exposed to corticosteroids [24, 25]. There have been concerns about additional cardiovascular risk factors after antenatal corticosteroid exposure, but there is limited evidence from randomized trials [26]. We therefore undertook a follow-up study of survivors whose mothers participated in the Auckland Steroid Trial, the first and therefore oldest randomized trial of antenatal corticosteroids [27]. We found that there were no clinically important differences between corticosteroid and placebo exposed participants at 50 years of age [28].

As part of that follow-up study, we showed that record linkage to routinely collected administrative data could not replace self-reported questionnaire data but rather, the two data sources were additive. Use of both sources increased case ascertainment and may therefore increase the power for detection of differences between randomized groups [29]. The primary analysis utilized adjudicated outcomes (any data source after adjudication by the expert panel), incorporating records from all data sources. However, in some studies, researchers have access to only one of the data sources to assess study outcomes. Therefore, this study aimed to determine whether the study findings would be altered if the outcomes were assessed by different data sources alone or in combination.

## Methods

The Auckland Steroid Trial, conducted in New Zealand from 1969 to 1974, was a randomized placebo controlled trial that aimed to prevent neonatal respiratory distress syndrome by administering antenatal betamethasone [27]. The trial was not registered as it was conducted before clinical trial registries were initiated. We traced the adult offspring of mothers who had been enrolled in the trial, requesting their participation through a self-reported questionnaire completion and consent to data linkage [28]. We obtained written consent from the participants in the study. The study was reviewed and approved by the Northern A Health and Disability Ethics Committee of New Zealand (IRB00008714). For this study, 03/03/2021 was the start date for obtaining the first self-reported questionnaire, and 31/05/2022 was the end date for receiving the last self-reported questionnaire. Final linked data were provided from various agencies between 04/04/2023 and 16/05/2023. Individuals provided informed written consent to data linkage and a unique national patient identifier for each participant was submitted to each agency providing the data linkage. This identifier was returned with the data so that it could be integrated with the other individual patient level data. Individuals could be identified by their national health information (NHI) number. The outcomes of interest were diabetes mellitus, pre-diabetes, total diabetes, hyperlipidemia, high blood pressure, mental health disorders, and asthma (S1 Appendix). We compared the relative risk of each outcome between the betamethasone and placebo groups, utilizing various data sources either individually or in combination to assess study outcomes.

### Data sources

The possible data sources for each outcome were the self-reported questionnaire data from the Auckland Steroid Trial follow-up study [28], linked administrative datasets (Table 1), both the questionnaire and administrative data (cases identified by both sources), either the questionnaire or administrative data (any data source), or adjudicated outcomes.

For all administrative datasets, the absence of any confirmatory evidence for a specific condition was assumed to represent no evidence for that condition. If discrepancies were observed

**Table 1. Data sources.**

| Dataset | Description | Extracted variables |
|---|---|---|
| National Minimum dataset (NMDS) | Public and private hospital discharge information, implemented in 1993 and backloaded with public hospital discharge information from 1988 | Hospital admissions with diagnostic codes for diabetes mellitus, hyperlipidemia, high blood pressure, depression or anxiety, and asthma. |
| Pharmaceutical Collection of the Ministry of Health | Pharmaceuticals dispensed in the community and funded by the New Zealand government, started in 1 July 1992 with complete *NHI links from 2006. | Records of prescription for diabetes mellitus, hyperlipidemia, high blood pressure, depression or anxiety, and asthma. |
| National Non-Admitted Patient Collection (NNPAC) | Data on outpatient and emergency department activity, implemented on 1 July 2006. | Records of attendance at a diabetes clinic or retinal screening for diabetic patients |
| Testsafe laboratory dataset | Laboratory test results from community laboratories in the Northern Region Health District. This region includes Auckland (New Zealand's largest city), where all participants were born, and which includes 36% of the New Zealand population. Of the trial participants, 63% reside in this region. Started in early 2010. | Haemoglobin A1c (HbA1c), plasma glucose concentration for diabetes. Total cholesterol, LDL, and triglyceride concentrations for hyperlipidemia. |
| Self-reported questionnaire | Included questions about chronic conditions, medical events, and mental health based on the New Zealand Health Survey [30]. | Participants were asked if they had ever been told by a doctor that they had specific diagnoses and what treatment they had received. |

*National Health Information

between the questionnaire and administrative data, or if evidence was available from a single administrative dataset, a consensus on the diagnosis was reached by an expert panel consisting of the five clinician members of the study steering group who reviewed all records including self-report data (adjudicated outcome) [29]. For direct comparability administrative data were right censored at date of completion of the self-reported questionnaire.

## Statistical analyses

We calculated descriptive statistics (mean (SD), n (%)) and also calculated adjusted relative risks (aRR) and 95% confidence intervals (CI) using generalized linear mixed modelling adjusting for sex and gestational age at trial entry [28] to fit a binomial distribution with a log link function for robust standard error estimates. We compared the relative risks from each data source to those assessed using adjudicated outcomes using the test of interaction [31]. Relative risks (95% CIs) are presented in forest plots. We also calculated the mean difference and the limits of agreement between relative risks derived from each data source to those from adjudicated outcomes using the Bland-Altman approach [32]. Data analysis was conducted using SAS (v9.4 SAS Institute Inc, Cary NC).

## Results

Of the 424 participants in the follow-up study (46% of those eligible), 415 (98%) completed a questionnaire, 420 (99%) consented to at least one administrative dataset, and 379 (89%) consented to all administrative data sources [29]. The mean age was 49 years (SD 1) in both the betamethasone (n = 229) and placebo groups (n = 195) (Table 2). The proportion of males was higher in the betamethasone group (124/229, 54.1%) than in the placebo group (88/195, 45.1%), but the proportion of preterm births, participants currently living overseas, consent for data linkage, questionnaire completion, and availability of administrative data were similar in both groups (Table 2). The self-reported questionnaire response rates were: 97% for diabetes, 98% for pre-diabetes, 97% for total diabetes, 96% for hyperlipidemia, 96% for high blood pressure, 96% for mental health disorders, and 97% for asthma.

**Table 2. Characteristics of participants whose mothers were randomized to betamethasone and placebo.**

| | Placebo | Betamethasone | *P value |
|---|---|---|---|
| | N = 195 | N = 229 | |
| | n (%) | n (%) | |
| Male | 88 (45.1) | 124 (54.1) | 0.06 |
| Age at follow up, years (mean, SD) | 49.3 (1.0) | 49.3 (1.0) | 1.00 |
| Preterm birth | 135 (69.2) | 166 (72.5) | 0.46 |
| Living in Northern geographic region | 108 (55.4) | 134 (58.5) | 0.51 |
| Living overseas | 16 (8.2) | 27 (11.8) | 0.22 |
| Always lived in New Zealand | 94 (48.2) | 107 (46.7) | 0.76 |
| Completed questionnaire | 192 (98.5) | 223 (97.4) | 0.44 |
| Consented to data linkage | 189 (96.9) | 223 (97.3) | 0.77 |
| Data available in TestSafe laboratory dataset | 138 (70.8) | 167 (73.0) | 0.62 |
| Data available in Pharmaceutical collection | 156 (80.0) | 190 (83.0) | 0.43 |

* Chi-square test or Student's t-test as appropriate.

Using adjudicated outcomes, there was no difference in risk between participants exposed to betamethasone and those exposed to placebo in the diagnosis of diabetes, pre-diabetes, total diabetes, hyperlipidemia, high blood pressure, mental health disorders, or asthma (Table 3, Figs 1 and 2).

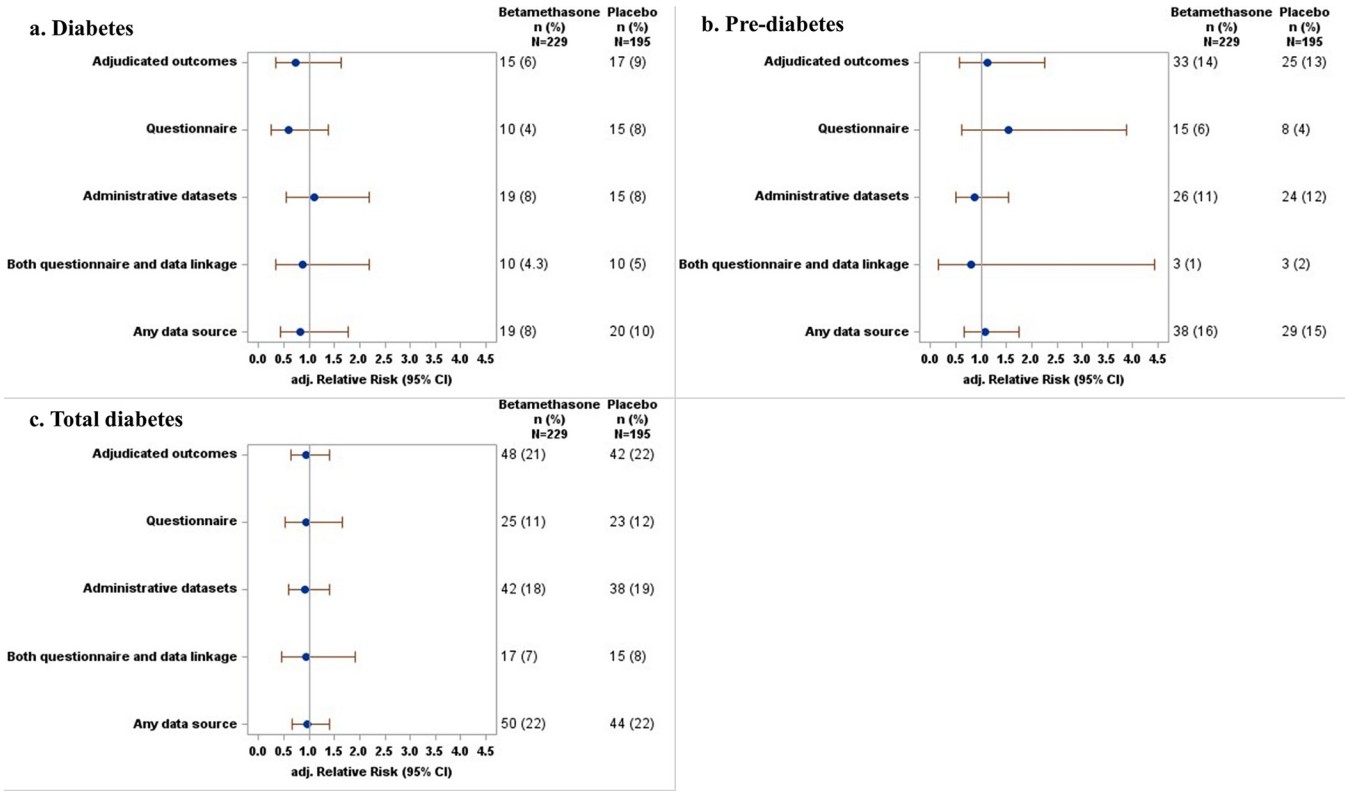

**Fig 1.** Forest plot of adjusted relative risks for comparison of the incidence of a. diabetes, b. pre-diabetes and c. total diabetes between betamethasone and placebo groups assessed using different data sources.

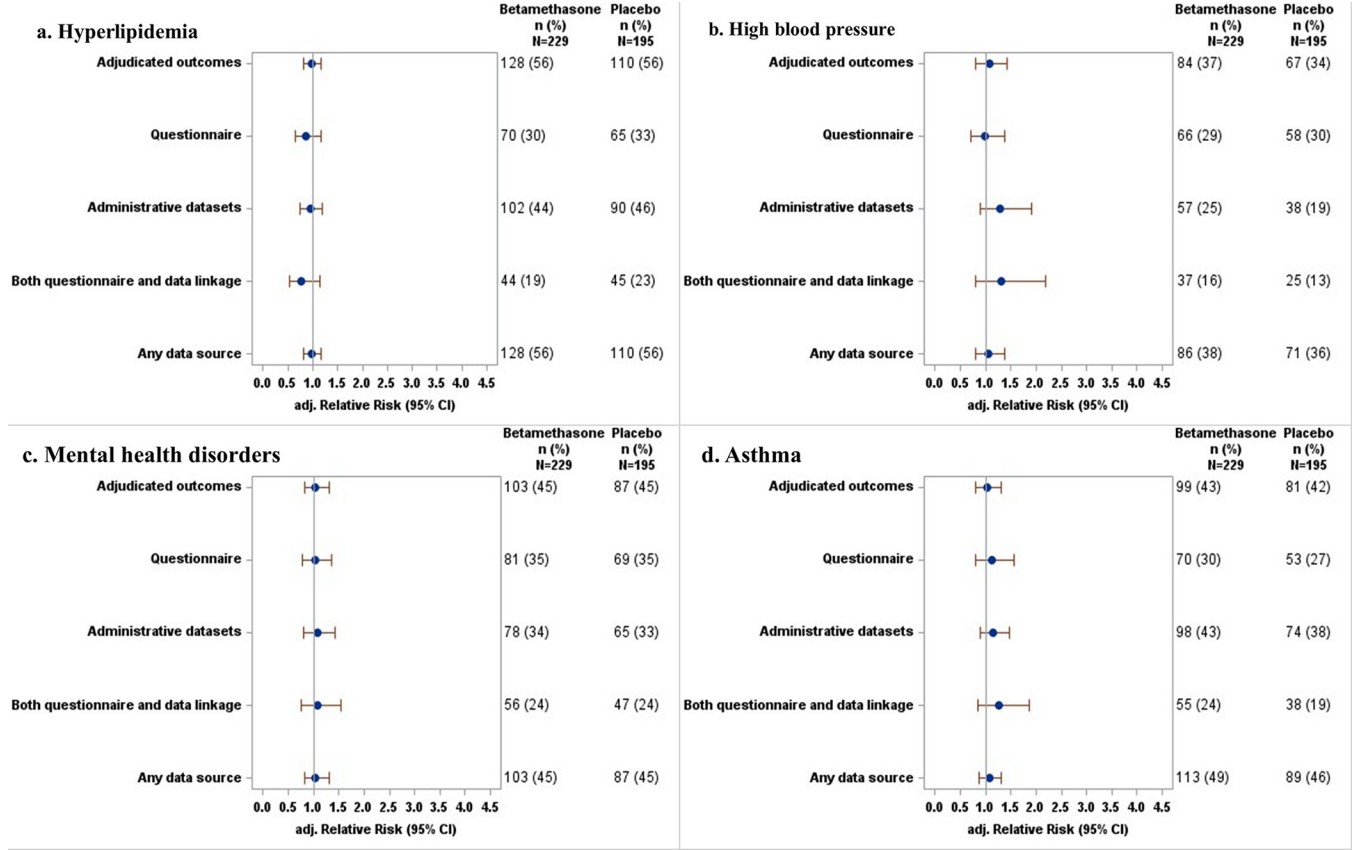

**Fig 2.** Forest plot of adjusted relative risks for comparison of the incidence of a. hyperlipidemia, b. high blood pressure, c. mental health disorders, and d. asthma between betamethasone and placebo groups assessed using different data sources.

There were also no differences in the outcomes between participants exposed to betamethasone and those exposed to placebo when outcomes were assessed using the self-reported questionnaire (mean difference in aRR 0.02, limits of agreement: -0.35 to 0.40, Table 3, Figs 1 and 2). The relative risks for the comparison between betamethasone and placebo groups for all outcomes assessed using adjudicated outcomes had similar magnitude to outcomes assessed using the self-reported questionnaire. The confidence intervals for estimates of treatment effects were slightly wider when using the self-reported questionnaire alone for all outcomes, except for diabetes where the confidence interval was narrower, compared to when using adjudicated outcomes (Table 3, Figs 1 and 2).

Similarly, the study outcomes did not differ between treatment groups when outcomes were assessed using administrative datasets (mean difference in aRR 0.06, limits of agreement: -0.32 to 0.44, Table 3, Figs 1 and 2). However, the risk of diabetes was non-significantly lower in the betamethasone group than in the placebo group when using adjudicated outcomes (aRR = 0.74, 95% CI [0.33, 1.64], P = 0.43), and non-significantly higher when using administrative datasets (aRR = 1.10, 95% CI [0.55, 2.18], P = 0.83), although confidence intervals showed substantial overlap (Fig 1, Table 3). Conversely, the risk of pre-diabetes was non-significantly higher in the betamethasone group than in the placebo group when using adjudicated outcomes (aRR = 1.13, 95% CI [0.56, 2.26], P = 0.67) but non- significantly lower when using administrative datasets (aRR = 0.87, 95% [CI 0.50, 1.53], P = 0.68), although again there was substantial overlap in the confidence intervals (Fig 1, Table 2). The confidence intervals

**Table 3. Study outcomes in the betamethasone and placebo groups estimated using each data source.**

| Outcome | Any data source after adjudication | | | Questionnaire data only | | | Administrative datasets only | | | Both questionnaire and administrative data. | | | Any data source | | |
|---|---|---|---|---|---|---|---|---|---|---|---|---|---|---|---|
| | Placebo group n (%) N = 195 | Betamethasone group n (%) N = 229 | aRR (95% CI), p-value | Placebo group n (%) N = 195 | Betamethasone group n (%) N = 229 | aRR (95% CI), p-value | Placebo group n (%) N = 195 | Betamethasone group n (%) N = 229 | aRR (95% CI), p-value | Placebo group n (%) N = 195 | Betamethasone group n (%) N = 229 | aRR (95% CI), p-value | Placebo group n (%) N = 195 | Betamethasone group n (%) N = 229 | aRR (95% CI), p-value |
| Diabetes | 17 (8.7) | 15 (6.5) | 0.74 (0.33, 1.64), 0.43 | 15 (7.6) | 10 (4.3) | 0.59 (0.25, 1.38), 0.17 | 15 (7.6) | 19 (8.2) | 1.10 (0.55, 2.18), 0.83 | 10 (5.1) | 10 (4.3) | 0.86 (0.33, 2.19), 0.72 | 20 (10.2) | 19 (8.2) | 0.83 (0.44, 1.77), 0.52 |
| Pre-diabetes | 25 (12.8) | 33 (14.4) | 1.13 (0.56, 2.26), 0.67 | 8 (4.1) | 15 (6.5) | 1.55 (0.62, 3.87), 0.29 | 24 (12.3) | 26 (11.3) | 0.87 (0.50, 1.53), 0.68 | 3 (1.5) | 3 (1.3) | 0.79 (0.14, 4.43), 0.64 | 29 (14.8) | 38 (16.5) | 1.07 (0.66, 1.74), 0.77 |
| Total diabetes (pre-diabetes or diabetes) | 42 (21.5) | 48 (20.9) | 0.94 (0.63, 1.40), 0.74 | 23 (11.7) | 25 (10.9) | 0.93 (0.52, 1.65), 0.79 | 38 (19.4) | 42 (18.3) | 0.91 (0.59, 1.39), 0.72 | 15 (7.6) | 17 (7.4) | 0.94 (0.46, 1.92), 0.92 | 44 (22.5) | 50 (21.8) | 0.96 (0.65, 1.41), 0.80 |
| Hyperlipidemia | 110 (56.4) | 128 (55.9) | 0.97 (0.81, 1.16), 0.82 | 65 (33.3) | 70 (30.5) | 0.87 (0.65, 1.16), 0.40 | 90 (46.1) | 102 (44.5) | 0.95 (0.75, 1.18), 0.69 | 45 (23.0) | 44 (19.2) | 0.77 (0.52, 1.14), 0.21 | 110 (56.4) | 128 (55.9) | 0.97 (0.81, 1.16), 0.82 |
| High blood pressure | 67 (34.3) | 84 (36.6) | 1.07 (0.81, 1.42), 0.61 | 58 (29.7) | 66 (28.8) | 0.99 (0.71, 1.37), 0.94 | 38 (19.4) | 57 (24.8) | 1.29 (0.88, 1.90), 0.20 | 25 (12.8) | 37 (16.1) | 1.30 (0.79, 2.18), 0.26 | 71 (36.4) | 86 (37.5) | 1.05 (0.80, 1.37), 0.69 |
| Mental health disorders | 87 (44.6) | 103 (44.9) | 1.04 (0.83, 1.30), 0.70 | 69 (35.3) | 81 (35.3) | 1.02 (0.78, 1.35), 0.87 | 65 (33.3) | 78 (34.0) | 1.07 (0.80, 1.42), 0.66 | 47 (24.1) | 56 (24.4) | 1.07 (0.75, 1.54), 0.71 | 87 (44.6) | 103 (44.9) | 1.04 (0.83, 1.30), 0.70 |
| Asthma | 81 (41.5) | 99 (43.2) | 1.02 (0.80, 1.30), 0.82 | 53 (27.1) | 70 (30.5) | 1.13 (0.81, 1.57), 0.45 | 74 (37.9) | 98 (42.7) | 1.14 (0.89, 1.46), 0.34 | 38 (19.4) | 55 (24.0) | 1.26 (0.84, 1.87), 0.22 | 89 (45.6) | 113 (49.3) | 1.07 (0.87, 1.30), 0.56 |

aRR, adjusted relative risk. CI, confidence interval.

were slightly wider when using administrative datasets alone for all outcomes, except for pre-diabetes where the confidence interval was narrower, compared to when using adjudicated outcomes (Table 3, Figs 1 and 2).

There was no difference in the outcomes between treatment groups when outcomes were assessed using both the questionnaire and administrative data (mean difference in aRR 0.01, limits of agreement: -0.41 to 0.43, Table 3, Figs 1 and 2). However, the risk of pre-diabetes was non-significantly higher in the betamethasone group than in the placebo group when using adjudicated outcomes (aRR = 1.13, 95% CI [0.56, 2.26], P = 0.67), but non-significantly lower and with wider CI when using both the questionnaire and administrative data (aRR = 0.79, 95% CI [0.14, 4.43], P = 0.64), although confidence intervals showed substantial overlap (Fig 1, Table 3).

The study outcomes did not differ between treatment groups when outcomes were assessed using any data source (mean difference in aRR 0.01, limits of agreement: -0.08 to 0.10, Table 3, Figs 1 and 2). The relative risks for comparison between betamethasone and placebo groups for all outcomes assessed using adjudicated outcomes had similar magnitude to outcomes assessed using any data source.

Comparison of aRRs assessed using questionnaire data, administrative datasets, both questionnaire and administrative data, or any data source with those calculated using adjudicated outcomes showed aRRs were not significantly different for all outcomes (Table 4). aRRs assessed using questionnaire data were also not different from those assessed using any administrative dataset.

## Discussion

We aimed to determine the effect of using different data sources alone or in combination, on the findings of a study of long-term effects of a randomized trial of antenatal corticosteroids. We found that there were no differences in the between group relative risk in any of study outcomes between those exposed to betamethasone and those exposed to placebo when outcomes were assessed using different data sources.

The findings of this 50-year follow-up study align with those of a previous 30-year follow-up study of the same cohort that found no differences between treatment groups in rates of

**Table 4. P-values for comparison of relative risks for study outcomes assessed using different data sources alone or in combination.**

|  | Questionnaire vs adjudicated outcomes P-value | Administrative datasets vs adjudicated outcomes P-value | Both questionnaire and administrative data vs adjudicated outcomes P-value | Any data source without adjudication vs adjudicated outcomes P-value | Questionnaire vs administrative datasets P-value |
|---|---|---|---|---|---|
| Diabetes | 0.70 | 0.46 | 0.81 | 0.83 | 0.26 |
| Pre-diabetes | 0.59 | 0.56 | 0.70 | 0.89 | 0.29 |
| Total diabetes | 0.97 | 0.92 | 0.99 | 0.94 | 0.95 |
| Hyperlipidemia | 0.53 | 0.88 | 0.29 | 0.99 | 0.63 |
| High blood pressure | 0.72 | 0.44 | 0.51 | 0.92 | 0.30 |
| Mental health disorders | 0.91 | 0.87 | 0.89 | 0.99 | 0.81 |
| Asthma | 0.62 | 0.52 | 0.37 | 0.76 | 0.96 |

P-values calculated using the test of interaction [31]

hypertension, systolic or diastolic blood pressure, diabetes mellitus, or fasting lipid concentrations [33]. It is also consistent with a 20-year follow-up of another randomized trial of antenatal corticosteroids that found no differences between treatment groups in the rate of mental health disorders [34].

The advantage of a self-reported questionnaire is that it is useful for participants living overseas, provides a definitive outcome in one step, requires no record linkage nor extraction of outcome data after that linkage, and is free from the associated costs and delays of record linkage and data extraction. Additionally self-reported questionnaires do not involve the participants in additional laboratory tests nor physical clinic visits. However, self-reporting non-severe conditions may be less reliable as participants might not be aware of their chronic conditions.

Administrative datasets used in our study included different types of information such as prescribed medications, laboratory test results, clinic attendance, and hospital admissions, making them reliable sources for outcome ascertainment. However, administrative data were not available for those living overseas, and laboratory results were only available for participants during the time they were residing in the northern geographic region. Given the advantages and disadvantages of each of these data sources, we found that either of them alone could be a useful source for outcome ascertainment, especially when data availability is not related to treatment assignment.

Using the self-reported questionnaire data alone yielded relative risks similar in magnitude and direction to those assessed using adjudicated outcomes for all study outcomes. Other studies have shown that patient-reported outcomes can serve as a useful tool for collecting outcome data [35, 36]. However, high rates of missing self-reported data can lead to an underestimation of outcomes and reduce the study's power to detect differences between treatment groups [37]. In our study the completion rate for self-reported questionnaires was high, and this was the sole method of obtaining data for participants living overseas. The questionnaire also identified more cases of high blood pressure and mental health disorders than administrative datasets where the only administrative data available were pharmaceutical dispensing records, although adding administrative data to the questionnaire data increased the number of cases for all outcomes [29]. For a severe outcome such as diabetes, participants may be more likely to be aware of their condition, leading to a lower likelihood of misclassification in self-reported data compared to administrative data and narrower confidence intervals for this outcome. While we have previously shown that the self-reported questionnaire underestimated all outcomes [29], this study confirmed that using the self-reported questionnaire alone to assess study outcomes would not alter the findings of the study on the long-term effects of antenatal corticosteroids on chronic conditions.

Similarly, using administrative data alone to assess differences between treatment groups revealed no significant differences in study outcomes, although the direction of relative risks for diabetes and pre-diabetes were reversed. This reversal appeared to occur because a small number of participants in one randomized group met the criteria for having the condition in administrative datasets, but were judged by the expert panel as being treated for other reasons (false positive) [29]. Despite this, the confidence intervals showed substantial overlap when using administrative datasets alone and when using adjudication outcomes to assess study outcomes. However, a higher rate of misclassifying outcomes may introduce bias in other studies. We have also previously shown that administrative data underestimated outcomes incidence, but using administrative datasets alone to assess outcomes had no impact on the result of the study when compared with adjudicated outcomes [29].

We found that the assessment of outcomes using a combination of questionnaire and administrative data gave similar estimates of treatment effects to those assessed using

adjudicated outcomes. A Cochrane systematic review of 47 randomized trials also reported that treatment effect estimates for outcome events assessed by adjudication committee did not differ from those assessed by onsite assessors [38]. Similarly, a study investigating the effects of randomized blood pressure lowering treatment on recurrent stroke using investigator diagnosis and adjudication by committee reported that the adjudication process had no apparent impact on the study's conclusion and argued that excluding adjudication could reduce the cost of conducting clinical studies [39]. Another study showed that routinely collected data could be used alone to assess serious vascular events in a follow-up study of myocardial infarction without the need for clinical adjudication [40]. Since we have previously reported that only a small number of participants with diabetes, pre-diabetes, high blood pressure, and asthma identified using the combined data sources were not confirmed by the expert panel, our findings align with other studies suggesting that adjudication may be unnecessary in determining the outcomes of a randomized trial [29].

Using a combination of questionnaire and administrative data can have several benefits for outcome ascertainment, potentially improving the study's power and helping to minimize underestimation of the outcomes. When there is a difference in the incidence of study outcomes between the treatment groups, underestimation of the outcomes, for example by using a single data source for ascertainment, could introduce bias either toward or away from the null [41]. In a meta-research study, which compared randomized trials using routinely collected data for outcome assessment versus traditional clinical trials, out of seven traditional clinical trials that reported statistically significant treatment benefits, three trials using routinely collected data showed no significant treatment benefit, three reported smaller treatment benefits (bias toward the null), and one showed a harmful effect of the treatment (bias away from the null) [10]. However, when there is no difference in the incidence of conditions between the treatment and placebo groups, as in our study, underestimation of the outcome using one data source may not significantly affect the study's results but could reduce the precision of the effect estimate. This reduction in precision was evident in our study, with slightly wider confidence intervals for estimates of treatment effects when using either the self-reported questionnaire or administrative datasets alone, as compared to using adjudicated outcomes.

## Strengths of the study

By utilizing both self-reported questionnaires and administrative data for outcome ascertainment, alongside high completion and consent rates for these data sources, our study was able to investigate the impact of utilizing these sources alone or in combination on trial findings. A high participation rate among those eligible for the study, coupled with unique identifiers for record linkage, were other strengths of our study.

## Limitations of the study

Our study lacked in-person clinical assessments, which could serve as the gold standard for comparisons. Our study may also involve some selection bias, given the relatively low follow-up rate. However, the baseline demographic variables of those who were eligible were similar to those of participants who consented, suggesting differential bias between randomised groups is unlikely.

Future studies encompassing a broad range of outcomes, including major events in a wider age range, and incorporating in-person assessments as a gold standard could provide greater insight into assessing potential underestimation of outcomes by various methods and the

impact of relying solely on self-reported questionnaires or administrative data in follow-up studies on the study findings.

## Conclusion

We aimed to investigate whether using different data sources to assess outcomes could impact the findings of a follow-up study of a randomized trial. We found that using a self-reported questionnaire alone, administrative datasets alone, a combination of both, or all data sources with or without adjudication, for assessing diabetes, pre-diabetes, total diabetes, hyperlipidemia, high blood pressure, mental health disorders and asthma in a follow-up study of a randomized trial had no impact on the study conclusions.

## Supporting information

**S1 Appendix. Outcome definitions [42, 43].**
(DOCX)

## Acknowledgments

We would like to express our gratitude to all the participants in this follow-up study.
ANCHOR Study Group: Jane E Harding (chair), Caroline A Crowther, Stuart R Dalziel, Carl L Eagleton, Greg D Gamble, Chris JD McKinlay.

## Author Contributions

**Conceptualization:** Jane E. Harding, Greg D. Gamble.

**Formal analysis:** Mohammad Shahbaz, Anthony Walters, Martin von Randow, Greg D. Gamble.

**Funding acquisition:** Jane E. Harding.

**Investigation:** Mohammad Shahbaz, Jane E. Harding, Barry Milne, Anthony Walters, Greg D. Gamble.

**Methodology:** Mohammad Shahbaz, Jane E. Harding, Greg D. Gamble.

**Resources:** Barry Milne.

**Supervision:** Jane E. Harding, Barry Milne, Greg D. Gamble.

**Validation:** Mohammad Shahbaz.

**Visualization:** Mohammad Shahbaz, Greg D. Gamble.

**Writing – original draft:** Mohammad Shahbaz.

**Writing – review & editing:** Mohammad Shahbaz, Jane E. Harding, Barry Milne, Anthony Walters, Martin von Randow, Greg D. Gamble.

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
