## [Decision Letter · Decision Letter 0]

3 Apr 2024

PONE-D-23-40655Effect of utilizing either a self-reported questionnaire or administrative data alone or in combination on the findings of a randomized controlled trial of the long-term effects of antenatal corticosteroids.PLOS ONE

Dear Dr. Gamble,

Thank you for submitting your manuscript to PLOS ONE. After careful consideration, we feel that it has merit but does not fully meet PLOS ONE’s publication criteria as it currently stands. Therefore, we invite you to submit a revised version of the manuscript that addresses the points raised during the review process.

We look forward to receiving your revised manuscript.

Kind regards,

Jayeshkumar Patel

Academic Editor

PLOS ONE

2. PLOS requires an ORCID iD for the corresponding author in Editorial Manager on papers submitted after December 6th, 2016. Please ensure that you have an ORCID iD and that it is validated in Editorial Manager. To do this, go to ‘Update my Information’ (in the upper left-hand corner of the main menu), and click on the Fetch/Validate link next to the ORCID field. This will take you to the ORCID site and allow you to create a new iD or authenticate a pre-existing iD in Editorial Manager. Please see the following video for instructions on linking an ORCID iD to your Editorial Manager account: https://www.youtube.com/watch?v=_xcclfuvtxQ".

4. Please remove your figures from within your manuscript file, leaving only the individual TIFF/EPS image files, uploaded separately. These will be automatically included in the reviewers’ PDF.

Additional Editor Comments:

Thank you for submitting your manuscript for consideration at PLOS One. Please review the comments from both reviewers and update the manuscript and/or results as necessary.

Reviewers' comments:

Reviewer's Responses to Questions

**Comments to the Author**

1. Is the manuscript technically sound, and do the data support the conclusions?

Reviewer #1: Yes

Reviewer #2: Yes

2. Has the statistical analysis been performed appropriately and rigorously? 

Reviewer #1: Yes

Reviewer #2: Yes

3. Have the authors made all data underlying the findings in their manuscript fully available?

Reviewer #1: No

Reviewer #2: No

4. Is the manuscript presented in an intelligible fashion and written in standard English?

Reviewer #1: Yes

Reviewer #2: Yes

5. Review Comments to the Author

Reviewer #1: The article investigates whether the use of self-reported questionnaires alone, administrative datasets alone, or a combination of both affects the study conclusions regarding chronic conditions among participants exposed to betamethasone versus placebo. While the study provides valuable insights into outcome ascertainment methods in clinical research and presents clear findings, there are several points that need to be addressed for the paper to be considered publishable.

#Method section:

• Please separate the data source section and section describing data sources and what variables was used from each one including the self-reported questionnaire.

• How missing data were handled or the potential implications of missing data would strengthen the analysis.

#Statistical analysis section:

• Please add a sentence about the descriptive statistics.

#Results section:

• Clarification is needed regarding the statistical test used to generate the p-value in table one.

#Discussion:

• Please explain the pros and cons of each method/data source and how can this have impacted your results. For example, administrative data for diabetes can miss cases that are not receiving glucose lowering medications.

• The paper mentions high completion rates for self-reported questionnaires. Could you please report the exact response rate in the results section by disease?

• The paper does not explicitly address potential selection bias.

• The discussion needs to highlight the limitations, strengths, and future direction.

Reviewer #2: The introduction lacks a clear explanation of the relationship between antenatal betamethasone and outcome variables. Given that the research explores differences between datasets concerning these variables, the rationale for studying this specific relationship warrants clarification. Is there prior research confirming a link between these variables? Furthermore, evidence of the long-term effects of antenatal betamethasone on outcome variables should be presented if it exists.

The discussion should connect the findings regarding betamethasone and outcome variables with existing literature.

In the results section, I suggest including the p-values alongside the adjusted relative risks and confidence intervals. Instead of presenting the data as 'aRR=1.13, 95% CI 0.56, 2.26', it should be 'aRR=1.13, 95% CI [0.56, 2.26], p-value = xx'.

The study frequently cites Shahbaz, M., et al., 'Comparison of outcomes of a randomized trial assessed by study questionnaire and by data linkage: the CONCUR study 2023,' yet the reference lacks a journal name, publication year, DOI, etc. This omission makes the study difficult to locate online.

The manuscript does not address the limitations inherent in self-reported questionnaires and administrative datasets, such as recall bias and the challenge of generalization.

The analysis accounted for two variables (sex and gestational age at the time of the trial). Conditions like diabetes, asthma, and hyperlipidemia are associated with numerous factors, including obesity, family history, and various other clinical and non-clinical factors. Were these factors unavailable in the dataset or not common in various datasets. Whatever the reason, authors should discuss the limitations of the analysis or model in the limitation section. A dedicated limitations section should be included at the end of the discussion.

The audience may not be as acquainted with the various administrative datasets employed in this study as the authors are. Hence, a concise description of these datasets and the key variables extracted is essential for reader comprehension.

6. PLOS authors have the option to publish the peer review history of their article (what does this mean?). If published, this will include your full peer review and any attached files.

Reviewer #1: **Yes: **Rowida Mohamed

Reviewer #2: No

---

## [Author Response · Author response to Decision Letter 0]

10 Jul 2024

As outlined in the Rebuttal letter we have tabulated Reviewers comments and out responses as in the rebuttal letter.

Comment Response

Reviewer 1

Please separate the data source section and section describing data sources and what variables was used from each one including the self-reported questionnaire. We added a data sources section subheading, and a table (Table 1) describing all datasets. 

How missing data were handled or the potential implications of missing data would strengthen the analysis. For all administrative datasets, the absence of any confirmatory evidence for a specific condition was assumed to represent no evidence for that condition. This is stated in the second paragraph of the “Data sources” section of the methods.

Please add a sentence about the descriptive statistics. This has been added to the first sentence of the “Statistical analysis” section.

Clarification is needed regarding the statistical test used to generate the p-value in table one. Added to the legend of table 2 (previously table 1). 

Please explain the pros and cons of each method/data source and how can this have impacted your results. For example, administrative data for diabetes can miss cases that are not receiving glucose lowering medications. We have added two paragraphs to the discussion section addressing these issues. (paragraph 3, the paragraph beginning “The advantage of a self-reported questionnaire” and paragraph 4, the paragraph beginning “Administrative datasets used in our”)

The paper mentions high completion rates for self-reported questionnaires. Could you please report the exact response rate in the results section by disease? The exact response rates have been added to the end of the first paragraph of the results section. (The self-reported questionnaire response rates were: 97% for diabetes, 98% for pre-diabetes, 97% for total diabetes, 96% for hyperlipidemia, 96% for high blood pressure, 96% for mental health disorders, and 97% for asthma.)

The paper does not explicitly address potential selection bias. Unrecognized selection bias may occur because eligible participants who moved overseas may be healthier, and those participants who did not consent to participation or who had died may have an increased burden of illness. However, the demographic characteristics of those who were eligible and those who consented to participation does not suggest any differential bias. We have added a sentence to the “Limitations” section of the discussion to address this.

The discussion needs to highlight the limitations, strengths, and future direction. Thanks for suggestion. All added at the end of discussion section.

Reviewer 2

The introduction lacks a clear explanation of the relationship between antenatal betamethasone and outcome variables. Given that the research explores differences between datasets concerning these variables, the rationale for studying this specific relationship warrants clarification. Is there prior research confirming a link between these variables? Furthermore, evidence of the long-term effects of antenatal betamethasone on outcome variables should be presented if it exists. An explanation has been added to paragraph 5 of the introduction.

The discussion should connect the findings regarding betamethasone and outcome variables with existing literature. Added as requested to the discussion section. (paragraph 2, the paragraph beginning “The findings of this 50-year follow-up study”).

In the results section, I suggest including the p-values alongside the adjusted relative risks and confidence intervals. Instead of presenting the data as 'aRR=1.13, 95% CI 0.56, 2.26', it should be 'aRR=1.13, 95% CI [0.56, 2.26], p-value = xx'. Thank you for the suggestion, the p-values were added to the Table 3 and the results section.

The study frequently cites Shahbaz, M., et al., 'Comparison of outcomes of a randomized trial assessed by study questionnaire and by data linkage: the CONCUR study 2023,' yet the reference lacks a journal name, publication year, DOI, etc. This omission makes the study difficult to locate online. Details have been added as requested.

The manuscript does not address the limitations inherent in self-reported questionnaires and administrative datasets, such as recall bias and the challenge of generalization. We have added limitations in using self-reported questionnaire in paragraph 3 of the introduction section. 

We have already noted the limitation of administrative datasets in paragraph 2 of the introduction section and the way to improve those limitations by combining with self-reported questionnaire in paragraph 4 and 6 of the introduction section. 

The analysis accounted for two variables (sex and gestational age at the time of the trial). Conditions like diabetes, asthma, and hyperlipidemia are associated with numerous factors, including obesity, family history, and various other clinical and non-clinical factors. Were these factors unavailable in the dataset or not common in various datasets. Whatever the reason, authors should discuss the limitations of the analysis or model in the limitation section. Because the study was the follow-up of a randomized clinical trial, and characteristics of participants were similar in both groups we have adopted a simple hypothesis testing approach rather than performing mediation analysis, since the exposure may drive intermediate steps and to adjust for them may discount any association between exposure and outcome. The aim of the current study was to determine whether self-reported outcome alone or in combination produced different findings in a trial context, not in relation to a mediation analysis. In the trial context covariates should be apparent at the time of randomization and post-randomization variables should not be include as confounders since they may be on the causal pathway to the outcomes of interest. 

A dedicated limitations section should be included at the end of the discussion. Added as requested. 

Please see response to reviewer 1 point.

The audience may not be as acquainted with the various administrative datasets employed in this study as the authors are. Hence, a concise description of these datasets and the key variables extracted is essential for reader comprehension. We have added a table describing all datasets. (Table 1) 

For extracted key variables please see the appendix 1.

---

## [Editor Report · Decision Letter 1]

24 Jul 2024

Effect of utilizing either a self-reported questionnaire or administrative data alone or in combination on the findings of a randomized controlled trial of the long-term effects of antenatal corticosteroids.

PONE-D-23-40655R1

Dear Dr. Gamble,

We’re pleased to inform you that your manuscript has been judged scientifically suitable for publication and will be formally accepted for publication once it meets all outstanding technical requirements.

Kind regards,

Jayeshkumar Patel

Academic Editor

PLOS ONE
---

## [Editor Report · Acceptance letter]

29 Jul 2024

PONE-D-23-40655R1 

PLOS ONE

Dear Dr. Gamble, 

I'm pleased to inform you that your manuscript has been deemed suitable for publication in PLOS ONE. Congratulations! Your manuscript is now being handed over to our production team.

Kind regards, 

on behalf of

Dr. Jayeshkumar Patel 

Academic Editor

PLOS ONE